# Reshaping the Regional Order of Health Care Resources in China: The Institutional Participants in an Inter-City Integrated Delivery System

**DOI:** 10.3390/ijerph18179176

**Published:** 2021-08-31

**Authors:** Xuanyi (Maxwell) Nie, Haobin (Bruce) Fan

**Affiliations:** 1Harvard T.H. Chan School of Public Health, Boston, MA 02115, USA; xnie@gsd.harvard.edu; 2Harvard University Graduate School of Design, Cambridge, MA 02138, USA; 3Shanghai Academy of Social Sciences, Shanghai 200020, China; 4School of Economics, Fudan University, Shanghai 200433, China; 5The Development Research Center of Shanghai Municipal People’s Government, Shanghai 200003, China

**Keywords:** integrated delivery system, organizational behavior, local governments, hospitals, regional governance

## Abstract

Over the past decades, pro-growth policies in China led to rapid economic development but overlooked the provision of health care services. Recently, increasing attention is paid to the emergence of integrated delivery systems (IDS) in China, which is envisioned to consolidate regional health care resources more effectively by facilitating patient referral among hospitals. IDS at an inter-city scale is particularly interesting because it involves both the local governments and the hospitals. Incentives among them will affect the development of an inter-city IDS. This paper thereby builds an economic model to examine both the inter-local government and inter-hospital incentives when participating in an inter-city IDS in China. The findings suggest that while inter-hospital incentives matter, inter-local government incentives should also be considered because the missing incentives at the local government level may oppose the development of inter-city IDSs.

## 1. Introduction

In contrast to other countries, China faces challenges in distributing health care services. About 80% of China’s health care resources and patients are concentrated in urban tertiary hospitals, while the remaining 20% are in primary hospitals or community-based general practice clinics, instead of the other way around. This “reverse pyramid” is widely criticized for its inefficiency in distributing health care resources [1]. With increasing awareness given to urban health care services for more human-centered governance, reforms have been made to improve the health care delivery system in Chinese cities. The 2009–2011 health care reform focused on the provision of universal health insurance coverage and affordable public health services [2,3]. The 2012-onwards reform strives to improve the delivery of health care in public hospitals [4]. This is accompanied by recent experiments with integrated delivery systems (IDS) that consolidate regional health care resources more effectively and redistribute patients through the referral network [5].

The central government of China has gradually released policy directives to encourage IDSs (the State Council of China published Guowei Yifa [2015] No. 70, stating that “medical resources should be shared to strengthen the medical capacity in primary care.” The former National Health and Family Planning Commission of China published the Guowei Yifa [2016] No. 75 to encourage urban tertiary hospitals to develop IDSs. The Guowei Yifa [2018] No. 26, published by the Chinese National Health Commission, aims for an evaluation system for IDSs), move patients between higher-level hospitals and lower-level health care providers. However, the results are not always successful [6]. One group of scholars focus on the lack of financial support from the government to covering the expanded insurance coverage and training of more health care professionals. The other group looks at inter-hospital conflicts such as administrative disorganization and difficulties in digitizing and sharing patient case files among hospitals. Understanding that the development of IDSs in China requires continuous efforts from various institutional stakeholders, therefore, it is necessary to probe the organizational logic among these institutional participants for better implementing IDSs.

However, the development of IDS in China is still at its nascent stage, which leads to a vacuum in literature. Only a few studies have discussed the organizational logic of IDSs. As a result, this paper argues that building an inter-city IDS requires an understanding of the inter-local government and inter-hospital relationships. This research thereby constructs an economic model to examine the incentives among these institutional partners and hypothesize that the missing incentives at the local government level may be an often-neglected factor that hinders the development of inter-city IDSs.

The paper is organized as such: the second section is a literature review on the definition of IDS and inter-city IDS, and current studies on the IDSs. The third section dives into the economic model, and the fourth section discusses the findings and implications, followed by the conclusions in the last section.

## 2. Materials and Methods

From a global perspective, an IDS is an integration of the delivery or provision of health care as opposed to a fragmented system. This practice is widely observed in the United Kingdom, United States, Australia, Japan, Germany, and Singapore [7]. In the United States, an IDS is a network of health care facilities under a parent holding company or a network of health care organizations constituting a corporate group that attempts to coordinate the patient journey across care transitions [8]. Based on the existing models, IDSs can be categorized into “vertical” and “horizontal” ones [9]. The “vertical” model integrates a continuum of services from prevention and diagnosis to rehabilitation and daily care. The referral usually happens among different levels of care providers. The “horizontal” model looks for one particular type of service with better quality. The referral is built by alliance, shareholding, or strategic partnership among health service providers at the same level.

In China, “vertical” IDSs usually occur at the county level, which, in the literature, is specifically described as yigongti. A county hospital takes the leadership to collaborate with lower-level county and village health care facilities, and integrates a spectrum of different types of care by referring patients to different levels of care providers [10]. On the contrary, “horizontal” IDSs are loosely defined. In literature, it is included under the general definition of yilianti, defined as a medical organization formed by different levels (provincial, municipal, county, village) of health care providers [11,12]. Patient referrals could occur among health care providers either at the same level or different levels.

Figure 1 illustrates the patient referrals for yilianti and yigongti between two cities. Both cities are depicted as reversed pyramids because health care resources are concentrated in tertiary hospitals. IDSs can be further divided into tight, semi-tight, and loose models [13]. A tight model refers to inter-hospital merging and acquisition, allowing the leading tertiary hospital to fully control the IDS. A semi-tight model allows the leading hospital to operate the IDS without changing the original asset holding structure. The loose model only shares the information, technology, equipment, and talents among entities in the IDS [14]. “Horizontal” IDSs could occur both at intra-city and inter-city scales. The Nanjing-Suqian IDS is an inter-city example [15]. The IDS is co-owned by the Nanjing Gulou Hospital (10% share), the Jinling Pharmaceutical Company Limited (63% share), and the Suqian city government (27% share). The Nanjing-Chuzhou IDS is another inter-city example across two provinces. The Chuzhou First People’s Hospital in Anhui province was purchased by the Nanjing Gulou Hospital in Jiangsu province (the announcement was made on the Website of Nanjing Gulou (Drum Tower) Hospital in 2016, retrieved from: https://www.njglyy.com/yydt/detail.asp?ID=2231, accessed on 26 July 2021) (see Figure 2). Figure 3 depicts the patient referral in an inter-city IDS. Both IDSs represent an innovative practice of regional coordination to reorganize health care resources (patient referral between the same-level hospitals).

As IDSs in the United States are usually the product of the corporate strategies of parent companies, research on the implementation of IDSs tends to focus on corporate organizational strategies [16]. Corporate leadership is identified as the most important factor in implementation [17,18], and research excludes the role of local governments, focusing on the health insurance coverage [19]. In China, most of the literature focuses on the “vertical” model, which was adopted to reform the hospital-centric and fragmented delivery system [20], and to “gatekeep” for health care service and health insurance [21,22,23]. Several studies have used economic theories to model the incentives and organizational logic in “vertical” IDSs. Liu and Dong [24] used the Stackelberg model to establish a game model between the leader hospital and the collaborating health care providers. Cui and Wang [25], using transaction economics theory, found that “tight” IDSs could reduce market transaction costs, while optimization strategies to reduce internal management transaction costs equally matter. Hospitals only collaborate when the expected outcome is larger than the cost [24,26]. Therefore, government policy directives are needed to reduce the transaction cost among hospitals for the development of IDSs [27].

However, these studies remained at the inter-hospital level, assuming that all levels of government are willing to invest in the development of IDSs. As experiments with the “horizontal” model came much later than these “vertical” models, no study has examined the organizational logics of a “horizontal” inter-city IDS. At the same time, the central government of China is actively promoting inter-city IDSs. The National Healthcare Security Administration [28] has been publishing policy directives to encourage “remote payment” (a patient pays for the treatments received in a city or province other than the city or province of the patient’s hukou residence), which is fundamental for inter-city patient referrals. Therefore, research on the inter-city IDS is significant to the current health care reform in China, and it is necessary to study the inter-local government relationship since reshaping regional health care resources is not only an inter-hospital game but also an inter-local government game.

## 3. Methods: The Economic Model

### 3.1. Inter-Local Government Level

The inter-local government relationship in China is theoretically competitive. The Chinese state has gone through neoliberal reforms and seen scaling down of governance from the central to the local governments, resulting in the so-called “local state corporatism” [29,30] or a “Chinese-style federalism” [31]. Health care resources are characterized as one of the “competitive public facilities” that pursue equity for the public, but also compete for market power and clients [32]. For local governments, establishing and maintaining the inter-city IDS is one cost. Another cost is the crowding-out effect. There is a threat of “free-ride” that patients from a less developed city are referred to the more developed city to receive the higher-quality health care service [33], occupying extra beds and crowding out local citizens from receiving care. Evidence shows that patients tend to concentrate in cities with more favorable quality of health care such as Beijing, Shanghai, and Guangzhou [34]. However, there are also gains for local governments. Displaying political adherence to central policy directives and achieving successful reform results could help a local official to be promoted [35]. Empirical studies also find that a local government tends to spend more on health care when its neighboring city increases its spending [36].

Table 1 summarizes the key variables to be considered in this model. The economic model assumes that the gain of a local government depends on the number of referred patients. This is a quantifiable measurement that could be promoted on news to demonstrate adherence to the central government policy. The gain is a function of two parameters: δ is the difference between medical service qualities in the referred-to city and the referred-from city, and τ as the crowding-out loss. The local governments determine the effort level e, which is measured by the public spending on the inter-city IDS. More public spending on the inter-city IDS indicates a more active response to the central policy. A model is set up to consider the effort levels of local governments in the inter-city IDS:
$$M=e^{2}\;{\left(\delta -\tau \right)}^{2},$$
C= τe,
where M is the political gain, and τ > 0 because the newly-referred patients increase the pressure on the referred-to city’s health care system. Although these referred patients may also spend on hotels, traveling, etc., thus making economic contributions to the referred-to city, these economic activities cannot instantly compensate for the loss in the health care system. So τ here is assumed greater than zero.

The cost C of implementing the inter-city IDS by a local government is represented by the crowding-out loss τ of the local patients due to the increased number of patients referred to the local hospitals, and is amplified by the effort level e. When δ > 0, it means that the quality of health care services is higher in the local city than the other cities, and vice versa, which could be measured by the Chinese hospital rating system. This rating system is the hospital qualification evaluation index based on hospital scale, scientific ability, personnel, technical strength, medical equipment, etc., and is applied to all hospitals regardless of hospital background, ownership, etc. According to the “Hospital Grading Management Standards”, hospitals can be classified into three levels, and each level is further divided into grade prime, grade middle, and grade low. For level three (the highest level), a super-prime grade is added. Thus, there are three levels and ten grades in total. $\frac{\partial M}{\partial e}>0$ indicates that the more effort a city spends on building the inter-city IDS, the more patients will be conveniently referred from the other city. The marginal cost of the effort level is held constant at $\frac{\partial C}{\partial e}=$τ. Each local government decides on its effort level at the margin, that is:$$\frac{\partial M}{\partial e}=\frac{\partial C}{\partial e},$$
$$2e(\delta -\tau {)}^{2}=\tau ,$$
and
$$e^{*}=\frac{\tau }{2(\delta -\tau {)}^{2}}.$$

The optimal level of effort for each local government depends on the difference in the quality of health care services and the crowding-out loss. Thus:$$\frac{\partial e^{*}}{\partial \delta }=-\tau (\delta -\tau {)}^{-3},$$
$$\frac{\partial e^{*}}{\partial \tau }=\frac{\delta^{2}-\tau^{2}}{2(\delta -\tau {)}^{4}}$$

The crowding-out loss parameter is a threshold to measure whether the quality of health care services in a city is better than the other city in the IDS. In general, when δ>τ, the higher-capacity and higher-service-quality hospital in a city could admit more referred-in patients without crowding out services for the local citizens. In contrast, if δ<τ, the quality of health care services provided by the hospital is inferior and the hospital is less capable of admitting referred patients.

Furthermore, when δ>τ, $\frac{\partial e^{*}}{\partial \delta }<0$ and $\frac{\partial e^{*}}{\partial \tau }>0$, but when δ<τ, $\frac{\partial e^{*}}{\partial \delta }>0$ and $\frac{\partial e^{*}}{\partial \tau }<0$. Between two cities in an inter-city IDS, assuming one city has the better quality of health care services. When the difference in the quality of health care service increases, the local government of this city is less willing to spend on the inter-city IDS. If the crowding-out loss increases, the local government spends more on the inter-city IDS. Assuming a city has the inferior quality of health care services, when the difference in the quality of health care service increases, the local government of this city is more willing to spend on the inter-city IDS. If the crowding-out loss increases, the local government spends less on the inter-city IDS. Although more patients are referred to a city with a higher quality of health care service and better hospitals, because the capacity of the referred-to hospital is usually very high (considering a more than 1000-bed mega hospital), and beds in higher-level urban hospitals are almost always fully occupied by both the local and non-local patients, the crowding-out loss τ could be assumed a constant.

The abovementioned relationship is illustrated in Figure 4. Given a certain value A for δ, the effort level of the referred-to city is higher than the effort level of the referred-from city. However, at the same time, the higher the value of A, the lower the value $e^{*}$ for both cities. That means the larger disparity between health care service qualities in two cities, the less both the two local governments are willing to spend on the inter-city IDS. The referred-to city is reluctant because patients are captured by its better quality of health care anyway, while the referred-from city is unwilling because its quality of health care service is already inferior, and the local government is more willing to spend on growth-oriented projects such as public infrastructures. Therefore, both of the two local governments are only willing to spend just enough to demonstrate adherence to the central government’s policies. This finding interestingly aligns with inter-local “collusion” in prior research on the political economy of China. O’Brien and Li [37] and Zhou [38] found that when confronting policy pressure from the central government, local governments in China tend to collude together to perfunctorily address policy arrangements to avoid unnecessary inter-local competition.

### 3.2. Inter-Hospital Level

The urban health insurances, or the Urban Employees Basic Medical Insurance (UEBMI) and the Urban Resident Basic Medical Insurance (URBMI), are financed by local governments and their residents [39]. At the same time, payment systems in China can be divided into the post-payment, or fee-for-service (FFS), and pre-payment systems that include various methods such as the global budget payment system (GBPS) and disease-related groups (DRGs) [40]. FFS can ensure the quality of care but can also lead to over-treatment and catastrophic expenditure [41]. On the contrary, pre-payment methods can control expenses by budgeting but may lead to decreased quality of care, and hospitals may tend to push away severely ill patients who need more treatments [42]. Currently, both FFS and GBPS exist in the Chinese health care system [43], and all levels of governments are working on replacing FFS with GBPS [44]. The economic model will examine both the FFS and GBPS systems.

Table 2 summarizes the key variables to be considered in the economic model. The inter-hospital relationship depends on the payment method. Health care providers are profit-driven, so they tend to compete to capture patients due to a zero-sum game [45]. As hospitals in China value efficiency more than equity, they further stress the zero-sum challenge [46]. Furthermore, the price p for both hospitals is assumed the same because prices were set by the government, especially for basic services to ensure affordable access to care. Providers received direct budgetary support to cover the difference between costs and revenues earned from these nominal fees [47]. In 2012, the Chinese government launched a nationwide reform, of which the key component was the zero-markup drug policy, which removed the previously allowed 15% markup for drug sales at public hospitals, and associated increases in fees for medical services [48]. Although regional (provincial) price difference exists, in 2020, the National Health Security Administration [49] made it clear that the current pricing system will be better centrally managed with a national standard.

#### 3.2.1. The Fee-For-Service (FFS)

Under FFS, referrals will make the referred-from hospital lose income because of fewer captured patients. When hospitals compete for patients, this is a leader-and-follower game. Assume that hospital 2 in city 2 is the leader of the inter-city IDS and hospital 1 in city 1 is the follower. By joining the IDS, hospitals have to decide how much effort they will make to facilitate the referrals. Hospital 1 and hospital 2 could respectively capture $q_{1}$ and $q_{2}$ number of patients, satisfying the following:$$q_{1}={\left(e_{1}+e_{2}\right)}^{2}-\gamma e_{2}{}^{2},$$
$$q_{2}={\left(e_{1}+e_{2}\right)}^{2}-\gamma e_{1}{}^{2}$$

In the above equations, $e_{i}$ denotes the effort level spent by hospital i, and γ ≥ 1 is the “discount factor” of effort made by the partner hospital. In the inter-city IDS, the leader hospital takes the initiative by demonstrating its effort, then the follower hospital observes this effort and decides on a certain level of effort as the best response. The leader hospital knows ex ante that the follower hospital observes its action and thus could commit to a Stackelberg leader action, in which the leader hospital’s best response would be to play the leader action.

Under FFS, where extra profits for hospitals are generated from capturing patients, the profits are:$$\pi_{1}=pq_{1}\left(e_{1},\;e_{2}\right)-c\left(q_{1},\;e_{1}\right),$$
$$\pi_{2}=pq_{2}\left(e_{1},\;e_{2}\right)-c\left(q_{2},\;e_{2}\right),$$
and p is the average treatment price for a patient per visit, and $c\left(q_{i},\;e_{i}\right)$ is the cost. The marginal cost of quantity is a constant $\frac{\partial c}{\partial q_{i}}=t$, the marginal cost of effort is stable at $\frac{\partial c}{\partial e_{i}}=m$, and $\frac{\partial^{2}c}{\partial q_{i}\partial e_{i}}=0$. In this Stackelberg model, the subgame perfect Nash equilibrium can be found using backward induction. Given the effort of the leader hospital $e_{2}$, the follower hospital’s best response satisfies the first condition:$$\frac{\partial \pi_{1}}{\partial e_{1}}=p\frac{\partial q_{1}}{\partial e_{1}}-\frac{\partial c}{\partial q_{1}}\frac{\partial q_{1}}{\partial e_{1}}-\frac{\partial c}{\partial e_{1}}=0,$$
$$2p\left(e_{1}+e_{2}\right)-2t\left(e_{1}+e_{2}\right)-m=0,$$
$$e_{1}^{*}=\frac{m}{2\left(p-t\right)}-e_{2}$$

When the leader hospital decides how much effort it contributes to the inter-city IDS, it takes into consideration the level of effort from the follower hospital. Therefore, the profit function is:$$\pi_{2}=pq_{2}\left(e_{1}\left(e_{2}\right),\;e_{2}\right)-c\left(q_{2},\;e_{2}\right).$$

So, the best response meets the following condition:$$\frac{\partial \pi_{2}}{\partial e_{2}}=p\left[\frac{\partial q_{2}}{\partial e_{1}}\frac{\partial e_{1}}{\partial e_{2}}+\frac{\partial q_{2}}{\partial e_{2}}\right]-\frac{\partial c}{\partial q_{2}}\frac{\partial q_{2}}{\partial e_{2}}-\frac{\partial c}{\partial e_{2}}=0,\;\left(p\gamma -t\right)e_{1}-te_{2}=\frac{m}{2}.$$

After substituting the best response of the follower hospital, the result is:$$\left(p\gamma -t\right)\left(\frac{m}{2\left(p-t\right)}-e_{2}\right)-te_{2}=\frac{m}{2},\;e_{2}^{*}=\frac{m}{2}\left(\frac{1-\frac{1}{\gamma }}{p-t}\right).$$

Thus, the best response of the follower hospital is:$$e_{1}^{*}=\frac{m}{2\left(p-t\right)\gamma }.$$

The “discount factor” γ denotes the potential or ability of a hospital to capture patients. It is positively correlated to the disparity between the qualities of health care service in the two hospitals. A higher value of γ implies that the disparity is high, so a patient is more likely to be captured by the other hospital in the inter-city IDS. From the model, it can be found that $\frac{\partial e_{1}^{*}}{\partial \gamma }<0$ and $\frac{\partial e_{2}^{*}}{\partial \gamma }>0$. Therefore, when γ is small, the leader hospital tends not to spend much effort to capture patients, while the follower hospital tends to spend effort competing. However, if γ is large, the leader hospital will spend more effort on building the inter-city IDS to capture more patients because the more convenient referral process brings more patients [1], while the follower hospital is unwilling to spend effort.

#### 3.2.2. The Global Budget Payment System (GBPS)

Under the GBPS, once the yearly budget is used up, the cost of treating extra patients is burdened by the hospital, so hospitals tend to push away patients [42,50,51]. When hospitals try to save their budgets, the inter-hospital relationship is a two-stage game. The number of patients captured by the hospitals,$\;q_{1}$ and $q_{2}$, remain the same to the Stackelberg model under the FFS. Under the GBPS, profits are generated from capturing patients within the predetermined budget constraint. Hospitals maximize their profits subject to their budget constraints by allocating portions of their yearly budgets as the quotas for the referred patients, given by:$$\max\;\pi_{1}=pq_{1}\left(e_{1},\;e_{2}\right)-c\left(q_{1},\;e_{1}\right)\;\;\;\;\;\;\;\;s.t.\;pq_{1}\left(e_{1},\;e_{2}\right)\leq E_{1},\;\max\;\pi_{2}=pq_{2}\left(e_{1},\;e_{2}\right)-c\left(q_{2},\;e_{2}\right)\;\;\;\;\;\;\;s.t.\;pq_{2}\left(e_{1},\;e_{2}\right)\leq E_{2}.$$

In these equations, $E_{1}$ is the predetermined budget quotas for patients referred to hospital 1, and $E_{2}$ is the predetermined budget quotas for patients referred to hospital 2; p is the average treatment price for a patient per visit, and $c\left(q_{i},\;e_{i}\right)$ is the cost. The marginal cost of quantity is a constant $\frac{\partial c}{\partial q_{i}}=t$, the marginal cost of effort is stable at $\frac{\partial c}{\partial e_{i}}=m$, and $\frac{\partial^{2}c}{\partial q_{i}\partial e_{i}}=0$. In this Stackelberg model, the subgame perfect Nash equilibrium can be found using the constraint optimization. Given the effort of the leader hospital $\;e_{2}$, for the follower hospital:$$\max\;L_{1}=pq_{1}\left(e_{1},\;e_{2}\right)-c\left(q_{1},\;e_{1}\right)-\lambda_{1}\left(pq_{1}\left(e_{1},\;e_{2}\right)-E_{1}\right),\;\frac{\partial L_{1}}{\partial e_{1}}=\left(1-\lambda_{1}\right)p\frac{\partial q_{1}}{\partial e_{1}}-\frac{\partial c}{\partial q_{1}}\frac{\partial q_{1}}{\partial e_{1}}-\frac{\partial c}{\partial e_{1}}=0,\;\frac{\partial L_{1}}{\partial e_{1}}=2\left(1-\lambda_{1}\right)p\left(e_{1}+e_{2}\right)-2t\left(e_{1}+e_{2}\right)-m=0,\;e_{1}^{*}=\frac{m}{2\left[\left(1-\lambda_{1}\right)p-t\right]}-e_{2}.$$

When the leader hospital decides how much effort it contributes to the inter-city IDS, it takes into consideration the effort level of the follower hospital. Therefore, for the leader hospital:$$\max\;L_{2}=pq_{2}\left(e_{1}\left(e_{2}\right),\;e_{2}\right)-c\left(q_{2},\;e_{2}\right)-\lambda_{2}\left(pq_{2}\left(e_{1}\left(e_{2}\right),\;e_{2}\right)-E_{2}\right).$$

Thus, the best response meets the condition given by:$$\frac{\partial L_{2}}{\partial e_{2}}=\left(1-\lambda_{2}\right)p\left[\frac{\partial q_{2}}{\partial e_{1}}\frac{\partial e_{1}}{\partial e_{2}}+\frac{\partial q_{2}}{\partial e_{2}}\right]-\frac{\partial c}{\partial q_{2}}\frac{\partial q_{2}}{\partial e_{2}}-\frac{\partial c}{\partial e_{2}}=0,\;\left[\left(1-\lambda_{2}\right)p\gamma -t\right]e_{1}-te_{2}=\frac{m}{2}.$$

Substituting the best response of the follower hospital into this equation, the result is:$$\left[\left(1-\lambda_{2}\right)p\gamma -t\right]\left[\frac{m}{2\left[\left(1-\lambda_{1}\right)p-t\right]}-e_{2}\right]-te_{2}=\frac{m}{2},\;e_{2}^{*}=\frac{m}{2\left[\left(1-\lambda_{1}\right)p-t\right]}\left[1-\frac{\left(1-\lambda_{1}\right)}{\left(1-\lambda_{2}\right)\gamma }\right].$$

Then, the best response of the follower hospital is:$$e_{1}^{*}=\frac{m}{2\left[\left(1-\lambda_{1}\right)p-t\right]}\frac{\left(1-\lambda_{1}\right)}{\left(1-\lambda_{2}\right)\gamma }.$$

The Lagrange multiplier $\lambda_{i}$ (i=1, 2) is the shadow price of the budget quota, which is the price that a hospital could afford for increasing the predetermined budget by one unit (one Chinese yuan), in order to increase the profit even more. In the model, since the predetermined budget is an inequality constraint, the Kuhn–Tucker conditions are adopted here and given by:$$E_{i}-pq_{i}\left(e_{1}^{*},e_{2}^{*}\right)\geq 0,\;\lambda_{i}^{*}\geq 0,\;\lambda_{i}^{*}\left[E_{i}-pq_{i}\left(e_{1}^{*},e_{2}^{*}\right)\right]=0.$$

Therefore, when $\lambda_{i}^{*}=0$, the constraint is slack, meaning hospital i is operating within the predetermined budget limit under GBPS. The optimal effort level of a hospital is similar to the one under FFS. Patients are willing to be referred to a city with better health care, which makes the “discount factor” γ larger. Thus, the leader hospital would spend more effort to capture more patients, while the follower hospital spends less effort, capturing fewer patients. When $\lambda_{i}^{*}>0$, the constraint is binding, meaning hospital i reaches its predetermined budget limit. The optimal effort levels depend on not just the “discount factor” γ, but also the shadow price of the budget quota. As hospitals have to bear the costs for treating patients after the budget quota is used up, they tend to push away patients. This further indicates that, if the predetermined budget quota is inflexible, the shadow price will be too high, driving the effort levels of both the leader and the follower hospitals to a very low level or even zero.

## 4. Results and Discussion

There are two major findings from the economic model. Firstly, there is a collusive relationship between the local governments in an inter-city IDS: both of them tend to spend minimally only enough to demonstrate adherence to the central policy directives. Secondly, the effort level of each hospital depends on the payment method. Under FFS, the leader hospital spends more effort to capture patients while the effort level of the follower hospital is low. Under GBPS, when the budget constraint is slack (such as at the start of the budgetary year), the relationship resembles the condition under FFS. When the budget constraint is binding (such as at the end of the budgetary year), the effort levels of both hospitals are low or even close to zero and tend to push away patients. The findings are characterized by effort levels in Table 3. It suggests that incentive incompatibilities could cause failure in building an inter-city IDS. While the inter-hospital relationship is one cause, the inter-local government relationship is equally important.

Furthermore, the government–hospital relationship could also affect the implementation of an inter-city IDS. The table suggests that only the leader hospital could have high incentives to push for the implementation of an inter-city IDS, and it needs a better patient referral system in an inter-city IDS to capture more patients for more revenue. However, the local governments are rather reluctant to invest in building the referral system, thereby compromising the interests of the leader hospital. This conflict provides for an explanation for the emergence of “tight” inter-city IDSs through inter-hospital mergers and acquisitions, as the local governments are unwilling to spend much effort building the IDS, the leader hospital has to act as a profit-driven corporation to purchase other hospitals to build their referral network, such as the ones in the Nanjing-Suqian and Nanjing-Chuzhou model [15]. Without hospitals’ initiative, the minimum level of local government efforts could only achieve loose inter-city IDSs.

More incentives should be provided for the local governments in building inter-city IDSs. Grants such as the “IDS budget fund” from the central government can relax the local government’s fiscal burden from the perspective of public spending. Furthermore, the results of building successful inter-city IDSs could be made a more upfront administrative evaluation criterion for local cadres. To ensure transparency and accountability, an evaluation agency could be established, preferably at the central level to avoid biased “self-evaluation” [52]. However, both the fiscal and administrative reforms require collaboration between multiple state agencies such as the National Health Commission, the Ministry of Finance, and the National Development and Reform Commission. This requires more profound institutional reforms. At the hospital level, incentives should be given to the leader hospital when the GBPS budget is tight. Provincial or national health care security administrations could be allowed to reimburse the extra cost generated by the referred patients admitted by the leader hospital, and the next-year budget could be adjusted accordingly. Or a more direct approach is to encourage inter-hospital M & A, exemplified by the “tight” IDS. Once the hospitals are merged under one parent company, financial conflicts can be internalized.

However, making inter-city IDSs work properly requires more research and experiments. While the local governments are reluctant to fund inter-city IDSs, the leader hospital already aspires to capture more patients, which could de facto lead to the concentration of patients and a high bed occupancy rate. However, one function of an IDS, particularly yigongti, is to gatekeep health care resources and health insurance. Although inter-city IDSs could mobilize and reorganize regional health care resources, the more profound question is, to what degree could this be achieved. For example, the IDS in the United States serves as a regional network of health care providers based on two-way referrals—patients can also be reversely referred from medical centers (hospitals) to sub-level rehabilitation hospitals. As for China, an urgent challenge is to develop this two-way referral in inter-city IDSs. While this research reveals the organizational logics of institutional participants in an inter-city IDS, it also reflects that, without an appropriate incentive design, patients tend to concentrate in the leader hospitals, exacerbating the already inefficient public hospital system 1. An inter-city IDS without effective gatekeeping would defy the purposes of IDS in China, which is to optimize the stymied public hospital system. Future research on the incentive design for referral gatekeeping is needed to understand the dynamics of implementing inter-city IDSs in China.

## 5. Conclusions

This paper studies an emerging health care phenomenon through the lens of organizational logic. Although IDSs have existed in many countries for decades, it is new to China and requires more substantial studies to understand the dynamics of implementing such systems. Existing studies focus on the “vertical” yigongti while not enough attention has been given to the “horizontal” referrals among the same-level health care providers. This research contributes to the current knowledge on IDSs by studying the “horizontal” referrals in an inter-city IDS and examining the roles of local governments. The economic model suggests interesting findings. Both the local governments in an inter-city IDS are not willing to spend effort building the IDS. The leader hospital of an inter-city IDS is always willing to spend effort capturing patients unless constrained by GBPS budget limits. The findings bring new knowledge to the current studies on IDSs in China by providing a perspective of inter-local government collusions. The economic model also explains the emergence of inter-hospital M & A when building “tight” IDSs, and the prevalence of “loose” IDSs because of minimal efforts from the local governments.

These findings suggest that further reforms on regional administrative and governance policies are needed to facilitate the development of inter-city IDSs. The significance of inter-city IDSs is the capacity to mobilize regional health care resources in response to the changing narrative of health care and demographic conditions in Chinese cities. However, the more profound question lies in the future direction of inter-city IDSs. Experiments with county-level yilianti have tested gatekeeping of IDSs, but are only limited to “vertical” models. Intra-city and inter-city IDSs have examined the potential of hospitals to reshape the regional order of health care resources but require further research on incentive design to establish gatekeeping. The findings are also contributive on the theoretical level. Firstly, our study broadens the scope of the current research on global health policies. By examining the organizational behaviors among the institutional actors, the theoretical model offers insights to the group of literature that attempts to understand health policies through an organizational perspective. Secondly, a majority of studies on the organization of the IDS are based on inter-hospital or hospital–society relationships, while our research offers an important understanding on the inter-local and government–hospital relationships due to the specific institutional structure in China. This brings further significance to the current theoretical works on health policy that, while some of the fundamental concepts can be shared across different regions, one health policy can have variegated forms of implementation, requiring scholars to situate the health policy in the broader socio-economic contexts.

Unavailability of data causes limitations for our study. Since this research is based on a deductive model, a gap between the findings of the economic model and the practical situation should be paid attention to. More studies in this area with sufficient data, particularly on the inter-hospital referrals, could employ empirical methods to test and substantiate our arguments. Future empirical research could investigate the impacts of inter-city IDSs on referred patients’ cost for health care and financial incentives or policy designs to gatekeep inter-city referral.

## Figures and Tables

**Figure 1 ijerph-18-09176-f001:**
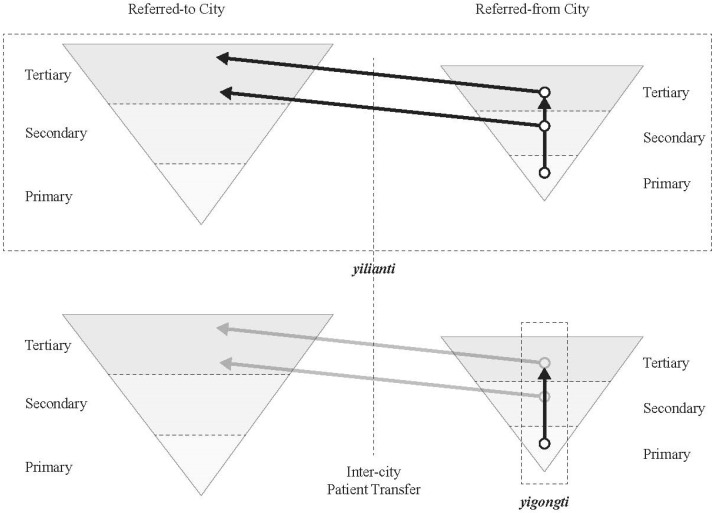
Patient referrals of yilianti and yigongti.

**Figure 2 ijerph-18-09176-f002:**
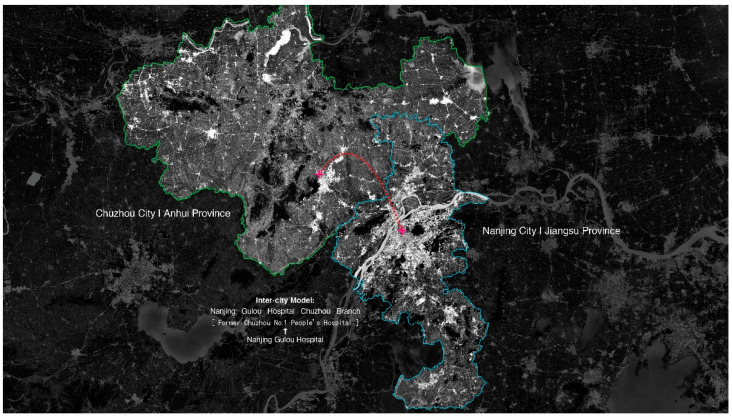
The Nanjing-Chuzhou inter-city’s (province) IDS.

**Figure 3 ijerph-18-09176-f003:**
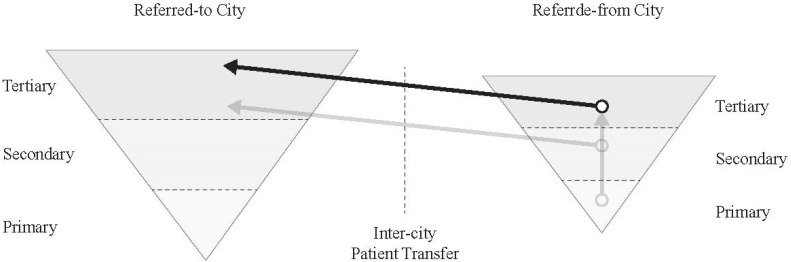
Patients’ transfer inside of an inter-city IDS.

**Figure 4 ijerph-18-09176-f004:**
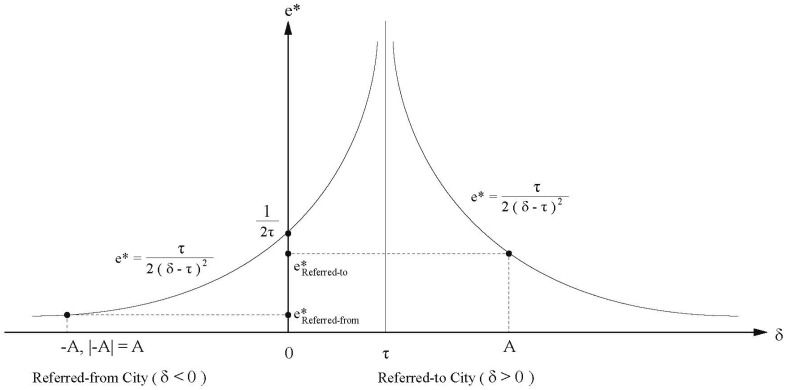
Algebraic relationship between δ (the differences of medical service quality between the referred-in city and the referred-from city) and $e^{*}$ (their optimal effort levels).

**Table 1 ijerph-18-09176-t001:** Key variables in the economic model at the inter-local government level.

Variables	Definition
δ	The difference between the quality of health care services of the two cities.
τ	The crowding-out loss of local citizens due to the increased number of referred patients to the city.
e	Effort level of both cities, measured by the level of public spending on the inter-city IDS.

**Table 2 ijerph-18-09176-t002:** Key variables in the conceptual model at the inter-hospital level.

Variables	Definition
$q_{1}$, $q_{2}$	In an inter-city IDS, hospital 1 in city 1 could capture $q_{1}$ patients referred from hospital 2 in city. 2. Hospital 2 in city 2 could capture $q_{2}$ patients referred from hospital 1 in city 1.
$e_{i}$, $\pi_{i}$	The effort level spent by hospital i, and the profit for hospital i.
γ≥1	The “discount factor” of effort made by the partner.
p	The average treatment price for a patient per visit, and $c\left(q_{i},\;e_{i}\right)$ is the cost.
$E_{1}$, $E_{2}$	The predetermined insurance budget quotas for hospital 1 and 2.

**Table 3 ijerph-18-09176-t003:** Summary of incentive compatibilities and incompatibilities.

	Incentives Represented by Effort Levels	
Local Governments:	**Referred-to City**	**Referred-from City**
	low	Low
Hospitals:	**Leader Hospital**	**Follower Hospital**
FFS:	high	low
GBPS:	high → low	low

## Data Availability

Not applicable.

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
