# Peer review of "Reshaping the Regional Order of Health Care Resources in China: The Institutional Participants in an Inter-City Integrated Delivery System"

_ijerph, 2021, doi:10.3390/ijerph18179176_

Round 1

Reviewer 1 Report

The effect of introducing new integrated delivery systems is verified through a deduction model. It is necessary to consider the following points:

First, there may be limitations in logical model construction in that it is based on a deductive model. It is necessary to organize this in the limitation section.

Second, there may be a gap between the theoretical model and the practical situation. It is necessary to describe this gap.

Third, what are the implications at the theoretical level? It is necessary to describe the aspects that contributed to the theories related to health policy.

Author Response

Thank you very much for reading our work and providing us with your insights. Please see our responses below per your comments:

1.First, there may be limitations in logical model construction in that it is based on a deductive model. It is necessary to organize this in the limitation section.

Thank you for pointing this out. Yes, one of our concerns is that this study is deduction-based research. Should we acquire a good dataset sufficient for inductive models, we would love to extend this study to another paper to substantiate some of the arguments we made. But for now, we will follow your suggestion and clarify this in the limitation, which is the last paragraph in the conclusion section (please read lines 439-443).

2.Second, there may be a gap between the theoretical model and the practical situation. It is necessary to describe this gap.

Thank you for this comment. We agree that there is a gap, but the gap is also primarily determined by the lack of data to conduct inductive models. Based on the reality of policy implementations, we could rely on our deductive model as of now to predict the behaviors of the institutional participants in an IDS. Combined with our responses to your comment#1, our future research aims at acquiring data and bridging this gap by comparing our findings in this study and the findings from the inductive models.

3.Third, what are the implications at the theoretical level? It is necessary to describe the aspects that contributed to the theories related to health policy.

Thank you for this comment. One of the original intentions of writing this research is to study the emerging practice of IDS in China based on current theories on health policies. We believe that at the theoretical level, our findings are contributive from three perspectives. Firstly, existing papers attempt to study the IDS at a global scale using organizational theories. We contribute to this group of literature by using the theory, which is elaborated by our economic model, to study the IDS in China, broadening the scope of the current literature. Secondly, a majority of studies on the organization of the IDS are based on inter-hospital or civic-hospital relations. We believe that understanding the IDS in China requires an examination of the inter-local and government-hospital relationships due to the institutional structure in China. This also brings further significance to the current theoretical works on health policy that, while some of the fundamental concepts such as the economic rationales and organizational logic can be shared across different regions, a health policy can have variegated forms when it is implemented. The IDS in China is an example.

Again, thank you very much for your time and patience spent with our research.

Reviewer 2 Report

The article concerns an interesting issue which is the economic analysis of the effectiveness of the integrated health care (IDS) model in China. The problem presented by the authors is all the more interesting as this concept originally comes from the free market health care system in the United States, while China represents a different model of health care.
The defined variables adopted in the study seem to correspond well to the presentation of the assumed concept. The methodology of the conducted research is correct from a scientific point of view. The conclusions presented in the paper correspond to the results of the study. One should agree with the authors' opinion that although the integrated medical care method (IDS) has been known for many years in developed countries (mainly in the United States but also in some European countries), this model is new in China. The more the obtained results may be valuable for the further development of the organization of this system.

Author Response

Comments: The article concerns an interesting issue which is the economic analysis of the effectiveness of the integrated health care (IDS) model in China. The problem presented by the authors is all the more interesting as this concept originally comes from the free-market health care system in the United States, while China represents a different model of health care.

The defined variables adopted in the study seem to correspond well to the presentation of the assumed concept. The methodology of the conducted research is correct from a scientific point of view. The conclusions presented in the paper correspond to the results of the study. One should agree with the authors' opinion that although the integrated medical care method (IDS) has been known for many years in developed countries (mainly in the United States but also in some European countries), this model is new in China. The more the obtained results may be valuable for the further development of the organization of this system.

Response: Thank you very much for your positive comments. We are glad that the emerging IDS in China is an interesting topic in economics, and hope our study could bring more attention to this field.